# Multi-Class Learning: From Theory to Algorithm

**Jian Li**[1,2]**, Yong Liu**[1]*, **Rong Yin**[1,2]**, Hua Zhang**[1]**, Lizhong Ding**[5]**, Weiping Wang**[1,3,4]
[1]Institute of Information Engineering, Chinese Academy of Sciences
[2]School of Cyber Security, University of Chinese Academy of Sciences
[3]National Engineering Research Center for Information Security
[4]National Engineering Laboratory for Information Security Technology
[5]Inception Institute of Artificial Intelligence (IIAI), Abu Dhabi, UAE
`{lijian9026,liuyong,yinrong,wangweiping}@iie.ac.cn`
`lizhong.ding@inceptioniai.org`

## Abstract

In this paper, we study the generalization performance of multi-class classification and obtain a shaper data-dependent generalization error bound with fast convergence rate, substantially improving the state-of-art bounds in the existing data-dependent generalization analysis. The theoretical analysis motivates us to devise two effective multi-class kernel learning algorithms with statistical guarantees. Experimental results show that our proposed methods can significantly outperform the existing multi-class classification methods.

## 1 Introduction

Multi-class classification is an important problem in various applications, such as natural language processing, information retrieval, computer vision, web advertising, etc. The statistical learning theory of binary classification is by now relatively well developed [19, 20, 21, 23, 27, 34], but there are still numerous statistical challenges to its multi-class extensions [25].

To understand the existing multi-class classification algorithms and guide the development of new ones, people have investigated the generalization ability of multi-class classification algorithms. In recent years, some generalization bounds have been proposed to estimate the ability of multi-class classification algorithms based on different measures, such as VC-dimension [1], Natarajan dimension [7], covering Number [9, 11, 37], Rademacher Complexity [5, 14, 27], Stability [10], PAC-Bayesian [26], etc. Although there have been several recent advances in the studying of generalization bounds of multi-class classification algorithms, convergence rates of the existing generalization bounds are usually $\mathcal{O}\big(K^2/\sqrt{n}\big)$, where $K$ and $n$ are the number of classes and size of the sample, respectively.

In this paper, we derive a novel data-dependent generalization bound for multi-class classification via the notion of local Rademacher complexity and further devise two effective multi-class kernel learning algorithms based on the above theoretical analysis. The rate of this bound is $\mathcal{O}\big((\log K)^{2+1/\log K}/n\big)$, which substantially improves on the existing data-dependent generalization bounds. Moreover, the proposed multi-class kernel learning algorithms have statistical guarantees and fast convergence rates. Experimental results on lots of benchmark datasets show that our proposed methods can significantly outperform the existing multi-class classification methods. The major contributions of this paper include: 1) A new local Rademacher complexity based bound with fast convergence rate for multi-class classification is established. Existing works [16, 27] for multi-class classifiers with Rademacher complexity does not take into account couplings among different classes. To obtain sharper bound, we introduce a new structural complexity result on function classes induced by general classes via the maximum operator, while allowing to preserve the correlations among

---

different components meanwhile. Thus, our result in this paper is a non-trivial extension of the binary classification of local Rademacher complexity to multi-classification; 2) Two novel multi-class classification algorithms are proposed with statistical guarantees: a) `Conv-MKL`. Using precomputed kernel matrices regularized by local Rademacher complexity, this method can be implemented by any $\ell_p$-norm multi-class MKL solvers; b) `SMSD-MKL`. This method puts local Rademacher complexity in penalized ERM with $\ell_{2,p}$-norm regularizer, implemented by stochastic sub-gradient descent with updating dual weights.

## 2 Related Work

### 2.1 Multi-Class Classification Bounds

**Rademacher Complexities Bounds.** Koltchinskii and Panchenko [14] and Koltchinskii, Panchenko, and Lozano [15] first introduced a margin-based bound for multi-class classification in terms of Rademacher complexity. This bound was slightly improved in [27, 5]. Maximov and Reshetova [25] gave a new Rademacher complexity based bound that is linear in the number of classes. Based on the $\ell_p$-norm regularization, Lei, Binder, and Klof [18] introduced a bound with a logarithmic dependence on the number of class size. Instead of global Rademacher complexity, in this paper, we use local Rademacher complexity to obtain a sharper bound, which substantially improves generalization performance upon existing global Rademacher complexity methods.

**VC-dimension Bounds.** Allwein, Schapire, and Singer [1] used the notion of VC-dimension for multi-class learning problems, and derived a VC-dimension based bound. Natarajan dimension was introduced in [28] in order to characterize multi-class PAC learnability, which exactly matches the notion of Vapnik-Chervonenkis dimension in the case of binary classification. Daniely and Shalev-Shwartz [7] derived a risk bound with Natarajan dimension for multi-class classification. VC dimension and Natarajan dimension are important tools to derive generalization bounds, however, these bounds are usually dimension dependent, which makes them hardly applicable to practical large-scale problems (such as typical computer vision problems).

**Covering Number Bounds.** Based on the $\ell_\infty$-norm covering number bound of linear operators, Guermeur [9] obtained a generalization bound exhibiting a linear dependence on the class size, which was improved by [37] to a radical dependence. Hill and Doucet [11] derived a class-size independent risk guarantee. However, their bound is based on a delicate definition of margin, which is not commonly used in mainstream multi-class literature.

**Stability Bounds and PAC-Bayesian Bounds.** Stability [10] and PAC-Bayesian [26] are two popular tools to analyze generalization performance on neural networks for multi-class setting. Hardt, Recht and Singer [10] generated generalization bounds for models learned with stochastic gradient descent. McAllester [26] proposed a dropout bound for neural networks with PAC-Bayesian. However, the convergence rate based on stability and PAC-Bayesian is usually at most $\mathcal{O}(1/\sqrt{n})$.

### 2.2 Local Rademacher Complexity

In recent years, several authors have applied *local* Rademacher complexity to obtain better generalization error bounds for traditional binary classification [2, 13, 22, 24], similar analysis has been explored in multi-label learning [35] and multi-task learning [36] as well. However, numerous statistical challenges remain in the multi-class case, and it is still unclear how to use this tool to derive a tighter bound for multi-class. In this paper, we bridge this gap by deriving a sharper generalization bound using local Rademacher complexity.

### 2.3 Multi-Class Kernel Learning Algorithms

As one of the success stories in multiple kernel learning, improvements in multi-class MKL have emerged [38], in which a one-stage multi-class MKL algorithm was presented as a generalization of multi-class loss function [6, 33]. And Orabona designed stochastic gradient methods, named OBSCURE [30] and UFO-MKL [29], which optimize primal versions of equivalent problems. In this paper, we consider the use of the local Rademacher complexity to devise the novel multi-class classification algorithms, which have statistical guarantees and fast convergence rates.

# 3 Notations and Preliminaries

We consider multi-class classification problems with $K \geq 2$ classes in this paper. Let $\mathcal{X}$ be the input space and $\mathcal{Y} = \{1, 2, \ldots, K\}$ the output space. Assume that we are given a sample $\mathcal{S} = \{z_1 = (\mathbf{x}_1, y_1), \ldots, z_n = (\mathbf{x}_n, y_n)\}$ of size $n$ drawn i.i.d. from a fixed, but unknown probability distribution $\mu$ on $\mathcal{Z} = \mathcal{X} \times \mathcal{Y}$. Based on the training examples $\mathcal{S}$, we wish to learn a scoring rule $h$ from a space $\mathcal{H}$ mapping from $\mathcal{Z}$ to $\mathbb{R}$ and use the mapping $\mathbf{x} \to \arg\max_{y \in \mathcal{Y}} h(\mathbf{x}, y)$ to predict. For any hypothesis $h \in \mathcal{H}$, the margin of a labeled example $z = (\mathbf{x}, y)$ is defined as

$$\rho_h(z) := h(\mathbf{x}, y) - \max_{y' \neq y} h(\mathbf{x}, y').$$

The $h$ misclassifies the labeled example $z = (\mathbf{x}, y)$ if $\rho_h(z) \leq 0$ and thus the expected risk incurred from using $h$ for prediction is $L(h) := \mathbb{E}_\mu[1_{\rho_h(z) \leq 0}]$, where $1_{t \leq 0}$ is the 0-1 loss, $1_{t \leq 0} = 1$ if $t \leq 0$, otherwise 0. Since 0-1 loss is hard to handle in learning machines, one usually considers the proxy loss: such as the square hinge $\ell(t) = (1 - t)_+^2$ and the square margin loss $\ell^s(t) = \left(1_{t \leq 0} + (1 - ts^{-1})1_{0 < t \leq s}\right)^2$, $s > 0$. In the following, we assume that: 1) $\ell(t)$ bounds the 0-1 loss: $1_{t \leq 0} \leq \ell(t)$; 2) $\ell$ is decreasing and it has a zero point $c_\ell$, i.e., $\ell(c_\ell) = 0$; 3) $\ell$ is $\zeta$-smooth, that is $|\ell'(t) - \ell'(s)| \leq \zeta|t - s|$. Note that both square hinge loss and margin loss satisfy the above assumptions.

Any function $h : \mathcal{X} \times \mathcal{Y} \to \mathbb{R}$ can be equivalently represented by the vector-valued function $(h_1, \ldots, h_K)$ with $h_j(\mathbf{x}) = h(\mathbf{x}, j), \forall j = 1, \ldots, K$. Let $\kappa : \mathcal{X} \times \mathcal{X} \to \mathbb{R}$ be a Mercer kernel with $\phi$ being the associated feature map, i.e., $\kappa(\mathbf{x}, \mathbf{x}') = \langle \phi(\mathbf{x}), \phi(\mathbf{x}') \rangle$. The $\ell_p$-norm hypothesis space associated with the kernel $\kappa$ is denoted by:

$$\mathcal{H}_{p,\kappa} = \left\{ h_{\mathbf{w}} = (\langle \mathbf{w}_1, \phi(\mathbf{x}) \rangle, \ldots, \langle \mathbf{w}_K, \phi(\mathbf{x}) \rangle) : \|\mathbf{w}\|_{2,p} \leq 1, 1 \leq p \leq 2 \right\}, \tag{1}$$

where $\mathbf{w} = (\mathbf{w}_1, \ldots, \mathbf{w}_K)$ and $\|\mathbf{w}\|_{2,p} = \left[\sum_{i=1}^K \|\mathbf{w}_i\|_2^p\right]^{\frac{1}{p}}$ is the $\ell_{2,p}$-norm. For any $p \geq 1$, let $q$ be the dual exponent of $p$ satisfying $1/p + 1/q = 1$.

The space of loss function associated with $\mathcal{H}_{p,\kappa}$ is denoted by

$$\mathcal{L} = \{\ell_h := \ell(\rho_h(z)) : h \in \mathcal{H}_{p,\kappa}\}. \tag{2}$$

Let $L(\ell_h)$ and $\hat{L}(\ell_h)$ be expected generalization error and empirical error with respect to $\ell_h$:

$$L(\ell_h) := \mathbb{E}_\mu[\ell(\rho_h(z))] \text{ and } \hat{L}(\ell_h) = \frac{1}{n} \sum_{i=1}^n \ell(\rho_h(z_i)).$$

**Definition 1** (Rademacher complexity). *Assume $\mathcal{L}$ is a space of loss functions as defined in Equation (2). Then the empirical Rademacher complexity of $\mathcal{L}$ is:*

$$\hat{\mathcal{R}}(\mathcal{L}) := \mathbb{E}_{\boldsymbol{\sigma}}\left[\sup_{\ell_h \in \mathcal{L}} \frac{1}{n} \sum_{i=1}^n \sigma_i \ell_h(z_i)\right],$$

*where $\sigma_1, \sigma_2, \ldots, \sigma_n$ is an i.i.d. family of Rademacher variables taking values -1 and 1 with equal probability independent of the sample $\mathcal{S} = (z_1, \ldots, z_n)$. The Rademacher complexity of $\mathcal{L}$ is $\mathcal{R}(\mathcal{L}) = \mathbb{E}_\mu \hat{\mathcal{R}}(\mathcal{L})$.*

Generalization bounds based on the notion of Rademacher complexity for multi-class classification are standard [14, 15, 27]: with probability $1 - \delta$, $L(h) \leq \inf_{0 < \gamma < 1} \left(\hat{L}(h_\gamma) + \mathcal{O}(\mathcal{R}(\mathcal{L})/\gamma + \log(1/\delta)/\sqrt{n})\right)$, where $\hat{L}(h_\gamma) = \frac{1}{n} \sum_{i=1}^n \left[1_{\rho_h(z_i) \leq \gamma}\right]$. Since $\mathcal{R}(\mathcal{L})$ is in the order of $\mathcal{O}(K^2/\sqrt{n})$ for various kernel multi-class in practice, so the standard Rademacher complexity bounds converge at rate $\mathcal{O}(K^2/\sqrt{n})$, usually.

Although Rademacher complexity is widely used in bound generalization analysis, it does not take into consideration the fact that, typically, the hypotheses selected by a learning algorithm have a better performance than in the worst case and belong to a more favorable sub-family of the set of all hypotheses [4]. Therefore, to derive sharper generalization bound, we consider the use of the local Rademacher complexity in this paper.

**Definition 2** (Local Rademacher Complexity). *For any $r > 0$, the local Rademacher complexity of $\mathcal{L}$ is defined as*

$$\mathcal{R}(\mathcal{L}^r) := \mathcal{R}\left\{ a\ell_h \Big| a \in [0,1], \ell_h \in \mathcal{L}, L[(a\ell_h)^2] \leq r \right\},$$

*where $L(\ell_h^2) = \mathbb{E}_\mu \left[ \ell^2(\rho_h(z)) \right]$.*

The key idea to obtain sharper generalization error bound is to choose a much smaller class $\mathcal{L}^r \subseteq \mathcal{L}$ with as small a variance as possible, while requiring that the solution is still in $\{h | h \in \mathcal{H}_{p,\kappa}, \ell_h \in \mathcal{L}^r\}$.

In the following, we assume that $\vartheta = \sup_{\mathbf{x} \in \mathcal{X}} \kappa(\mathbf{x}, \mathbf{x}) < \infty$, and $\ell_h : \mathcal{Z} \to [0, d]$, $d > 0$ is a constant. The above two assumptions are two common restrictions on kernel function and loss functions, which are satisfied by the popular Gaussian kernels and the bounded hypothesis, respectively.

## 4 Sharper Generalization Bounds

In this section, we first estimate the local Rademacher complexity, and further derive a sharper generalization bound.

### 4.1 Local Rademacher Complexity

The estimate the local Rademacher complexity of multi-class classification is given as follows.

**Theorem 1.** *With probability at least $1 - \delta$,*

$$\mathcal{R}(\mathcal{L}^r) \leq \frac{c_{d,\vartheta}\xi(K)\sqrt{\zeta r}\log^{\frac{3}{2}}(n)}{\sqrt{n}} + \frac{4\log(1/\delta)}{n},$$

*where*

$$\xi(K) = \begin{cases} \sqrt{e}(4\log K)^{1 + \frac{1}{2\log K}}, & \text{if } q \geq 2\log K, \\ (2q)^{1 + \frac{1}{q}} K^{\frac{1}{q}}, & \text{otherwise,} \end{cases}$$

*$c_{d,\vartheta}$ is a constant depends on $d$ and $\vartheta$.*

Note that the order of the (global) Rademacher complexity over $\mathcal{L}$ is usually $\mathcal{O}\big(K^2/\sqrt{n}\big)$ for various kernel multi-classes. From Theorem 1, one can see that the order of the local Rademacher complexity is $\mathcal{R}(\mathcal{L}^r) = \mathcal{O}\big(\sqrt{r}\xi(K)/\sqrt{n} + 1/n\big)$. Note that $\xi(K)$ is logarithmic dependence on $K$ when $q \geq 2\log K$. For $2 \leq q < 2\log K$, $\xi(K) = \mathcal{O}(K^{\frac{2}{q}})$ which is also substantially milder than the quadratic dependence for Rademacher complexity. If we choose a suitable value of $r$, the order can even reach $\mathcal{O}\big((\log K)^{2 + 1/\log K}/n\big)$ (see in the next subsection), which substantially improves the Rademacher complexity bounds.

### 4.2 A Sharper Generalization Bound

A sharper bound for multi-class classification based on the notion of local Rademacher complexity is derived as follows.

**Theorem 2.** *$\forall h \in \mathcal{H}_{p,\kappa}$ and $\forall k > \max(1, \frac{\sqrt{2}}{2d})$, with probability at least $1 - \delta$, we have*

$$L(h) \leq \max\left\{ \frac{k}{k-1}\hat{L}(\ell_h), \hat{L}(\ell_h) + \frac{c_{d,\vartheta,\zeta,k}\xi^2(K)\log^3 n}{n} + \frac{c_\delta}{n} \right\},$$

*where*

$$\xi(K) = \begin{cases} \sqrt{e}(4\log K)^{1 + \frac{1}{2\log K}}, & \text{if } q \geq 2\log K, \\ (2q)^{1 + \frac{1}{q}} K^{\frac{1}{q}}, & \text{otherwise,} \end{cases}$$

*$c_{d,\vartheta}$ is a constant depending on $d, \vartheta, \zeta, k$, and $c_\delta$ is a constant depending on $\delta$.*

The order of the generalization bound in Theorem 2 is $\mathcal{O}\big(\xi^2(K)/n\big)$. From the definition of $\xi(K)$, we can obtain that

$$
\mathcal{O}\left(\frac{\xi^2(K)}{n}\right) = \begin{cases} \mathcal{O}\left((\log K)^{2+1/\log K}/n\right), & \text{if } q \geq 2\log K, \\ \mathcal{O}\left(K^{2/q}/n\right), & \text{if } 2 \leq q < 2\log K. \end{cases}
$$

Note that our bounds is linear dependence on the reciprocal of sample size $n$, while for the existing data-dependent bounds are all radical dependence. Furthermore, our bounds enjoy a mild dependence on the number of classes. The dependence is polynomial with degree $2/q$ for $2 \leq q < 2\log K$ and becomes logarithmic if $q \geq 2\log K$, which is substantially milder than the quadratic dependence established in [14, 15, 27, 5].

### 4.3 Comparison with the Related Work

**Rademacher Complexity Bounds** Koltchinskii and Panchenko [14] and Koltchinskii, Panchenko, and Lozano [15] introduce a margin-based bound for multi-class classification in terms of Rademacher complexities: $L(h) \leq \inf_{0<\gamma<1} \hat{L}(h_\gamma) + \mathcal{O}\big(\frac{K^2}{\gamma\sqrt{n}} + \frac{\log 1/\delta}{\sqrt{n}}\big)$. The order is $\mathcal{O}\big(\frac{K^2}{\sqrt{n}}\big)$, which is slightly improved (by a constant factor prior to the Rademacher complexity term) by [27, 5]. Maximov and Reshetova [25] give a new Rademacher complexity bound: $L(h) \leq \inf_{0<\gamma<1} \hat{L}(h_\gamma) + \mathcal{O}\big(K/(\gamma\sqrt{n}) + \log(1/\delta)/\sqrt{n}\big)$, which has the form of $\mathcal{O}\big(K/\sqrt{n}\big)$. Based on the $\ell_p$-norm regularization, Lei, Binder, and Klof [18] derive a new bound: $L(h) \leq \hat{L}(\ell_h) + \mathcal{O}\big(\log^2 K/\sqrt{n}\big)$. The existing bounds based on Rademacher complexity are all radical dependence on the reciprocal of sample size.

In this paper, we derive a sharper bound based on the local Rademacher complexity with order $\mathcal{O}\big((\log K)^{2+\frac{1}{\log K}}/n\big)$, substantially sharper than the existing bounds of Rademacher complexity.

**Covering Number Bounds** Based on the $\ell_\infty$-norm covering number bound of linear operators, Guermeur [9] obtains a generalization of form $\mathcal{O}\big(K/\sqrt{n}\big)$, which is improved by [37] to a radical dependence: $L(h) \leq \hat{L}(\ell_h) + \mathcal{O}\big(\sqrt{K/n}\big)$. Hill and Doucet [11] derive a class-size independent risk guarantee of form $\mathcal{O}\big(\sqrt{1/n}\big)$. However, their bound is based on a delicate definition of margin, which is not commonly used in mainstream multi-class literature.

**VC-dimension Bounds** VC-dimension is an important tool to derive the generalization bound for binary classification. Allwein, Schapire, and Singer [1] show how to use it for multi-class learning problems, and derive a VC-dimension based bounds: $L(h) \leq \hat{L}(h_\gamma) + \mathcal{O}\big(\sqrt{V}\log K/\sqrt{n}\big)$, where $V$ is the VC-dimension. Natarajan dimension is introduced in [28] in order to characterize multi-class PAC learnability. Daniely and Shalev-Shwartz [7] derive a generalization bound with Natarajan dimension: $L(h) \leq \hat{L}(h_\gamma) + \mathcal{O}\big(d_{Nat}/n\big)$, where $d_{Nat}$ is the Natarajan dimension. Note that VC dimension bounds, as well as Natarajan dimension bounds, are usually dimension dependent, which makes them hardly applicable for practical large scale problems (such as typical computer vision problems).

**Stability and PAC-Bayesian Bounds** Stability [10] and PAC-Bayesian [26] are two useful tools to analyze generalization performance on neural networks for a multi-class setting. Hardt, Recht and Singer [10] generated generalization bounds for models learned with stochastic gradient descent using stability: $L(h) \leq \hat{L}(h_\gamma) + \mathcal{O}\big(1/\sqrt{n}\big)$. McAllester [26] used the PAC-Bayesian theory to derive generalization bound: $L(h) \leq \hat{L}(h_\gamma) + \mathcal{O}\big(\sqrt{\hat{L}(h_\gamma)/n}\big)$.

## 5 Multi-Class Multiple Kernel Learning

Motivated by the above analysis of generalization bound, we will exploit the properties of the local Rademacher complexity to devise two algorithms for multi-class multiple kernel learning (MC-MKL).

In this paper, we consider the use of multiple kernels, $\kappa_{\boldsymbol{\mu}} = \sum_{m=1}^M \mu_m \kappa_m$. A common approach to multi-class classification is the use of joint feature maps $\phi(\mathbf{x}) : \mathcal{X} \to \mathcal{H}$ [33]. For multiple kernel learning, we have $M$ feature mappings $\phi_m$, $m = 1, \ldots, M$ and $\kappa_m(\mathbf{x}, \mathbf{x}') = \langle \phi_m(\mathbf{x}), \phi_m(\mathbf{x}') \rangle$, where $m = 1, \ldots, M$. Let $\phi_{\boldsymbol{\mu}}(\mathbf{x}) = [\phi_1(\mathbf{x}), \ldots, \phi_M(\mathbf{x})]$. Using Theorem 2, to obtain a shaper

generalization bound, we confine $q \geq 2\log K$, thus $1 < p \leq \frac{2\log K}{2\log K-1}$. The $\ell_p$ hypothesis space of multiple kernels can be written as:

$$\mathcal{H}_{mkl} = \left\{ h_{\mathbf{w},\kappa_{\boldsymbol{\mu}}} = (\langle \mathbf{w}_1, \phi_{\boldsymbol{\mu}}(\mathbf{x}) \rangle, \ldots, \langle \mathbf{w}_K, \phi_{\boldsymbol{\mu}}(\mathbf{x}) \rangle), \|\mathbf{w}\|_{2,p} \leq 1, 1 < p \leq \frac{2\log K}{2\log K - 1} \right\}.$$

## 5.1 Conv-MKL

The global Rademacher complexity of $\mathcal{H}_{mkl}$ can be bounded by the trace of kernel matrix $\mathbf{K}_{\boldsymbol{\mu}} = \sum_{m=1}^{M} \mathbf{K}_m$. Existing works on [17, 32] use the following constraint to $\mathcal{H}_{mkl}$: $\mathrm{Tr}(\mathbf{K}_{\boldsymbol{\mu}}) \leq 1$. According to the above theoretical analysis, the local Rademacher complexity (the tail sum of the eigenvalues of the kernel) leads to tighter generalization bounds than the global Rademacher complexity (the trace). Thus, we add the local Rademacher complexity to restrict $\mathcal{H}_{mkl}$:

$$\mathcal{H}_1 = \left\{ h_{\mathbf{w},\kappa_{\boldsymbol{\mu}}} \in \mathcal{H}_{mkl} : \sum_{j>\zeta} \lambda_j(\mathbf{K}_{\boldsymbol{\mu}}) \leq 1 \right\},$$

where $\lambda_j(\mathbf{K}_{\boldsymbol{\mu}})$ is the $j$-th eigenvalues of $\mathbf{K}_{\boldsymbol{\mu}}$ and $\zeta$ is free parameter removing the $\zeta$ largest eigenvalues to control the tail sum. Note that the tail sum is the difference between the trace and the $\zeta$ largest eigenvalues: $\sum_{j>\zeta} \lambda_j(\mathbf{K}_{\boldsymbol{\mu}}) = \mathrm{Tr}(\mathbf{K}_{\boldsymbol{\mu}}) - \sum_{j=1}^{\zeta} \lambda_j(\mathbf{K}_{\boldsymbol{\mu}})$, thus the tail sum can be calculated in $O(n^2\zeta)$ for each kernel.

One can see that $\mathcal{H}_1$ is not convex, and we know that: $\sum_{m=1}^{M} \mu_m \sum_{j>\zeta} \lambda_j(\mathbf{K}_m) = \sum_{m=1}^{M} \mu_m/\|\boldsymbol{\mu}\|_1 \sum_{j>\zeta} \lambda_j(\|\boldsymbol{\mu}\|_1 \mathbf{K}_m) \leq \sum_{j>\zeta} \lambda_j(\mathbf{K}_{\boldsymbol{\mu}})$. Thus, we consider the use of the convex $\mathcal{H}_2$:

$$\mathcal{H}_2 = \left\{ h_{\mathbf{w},\kappa_{\boldsymbol{\mu}}} \in \mathcal{H}_{mkl} : \sum_{m=1}^{M} \mu_m \sum_{j>\zeta} \lambda_j(\mathbf{K}_m) \leq 1 \right\}.$$

According to normalized kernels $\tilde{\kappa}_m = \left( \sum_{j>\zeta} \lambda_j(\mathbf{K}_m) \right)^{-1} \kappa_m$ and $\tilde{\kappa}_{\boldsymbol{\mu}} = \sum_{m=1}^{M} \mu_m \tilde{\kappa}_m$, we can simply rewrite $\mathcal{H}_2$ as $\left\{ h_{\mathbf{w},\tilde{\kappa}_{\boldsymbol{\mu}}} = \left( \langle \mathbf{w}_1, \tilde{\phi}_{\boldsymbol{\mu}}(\mathbf{x}) \rangle, \ldots, \langle \mathbf{w}_K, \tilde{\phi}_{\boldsymbol{\mu}}(\mathbf{x}) \rangle \right), \|\mathbf{w}\|_{2,p} \leq 1, 1 < p \leq \frac{2\log K}{2\log K - 1}, \boldsymbol{\mu} \succeq 0, \|\boldsymbol{\mu}\|_1 \leq 1 \right\}$, which is a commonly studied hypothesis class in multi-class multiple kernel learning. A simple process with precomputed kernel matrices regularized by local Rademacher complexity can be seen in Algorithm 1:

---

**Algorithm 1** Conv-MKL

---

**Input:** precomputed kernel matrices $\mathbf{K}_1, \ldots, \mathbf{K}_M$ and $\zeta$
**for** $i = 1$ **to** $M$ **do**
    Compute tail sum: $r_m = \sum_{j>\zeta} \lambda_j(\mathbf{K}_m)$
    Normalize precomputed kernel matrix: $\widehat{\mathbf{K}}_m = \mathbf{K}_m/r_m$
**end for**
Use $\widehat{\mathbf{K}}_m, m = 1, \ldots, M$, as the basic kernels in any $\ell_p$-norm MKL solver

---

## 5.2 SMSD-MKL

Considering a more challenging case, we perform penalized ERM over the class $\mathcal{H}_1$, aiming to solve a convex optimization problem with an additional term representing local Rademacher complexity :

$$\min_{\mathbf{w},\boldsymbol{\mu}} \underbrace{\frac{1}{n} \sum_{i=1}^{n} \ell(\mathbf{w}, \phi_{\boldsymbol{\mu}}(\mathbf{x}_i), y_i)}_{C(\mathbf{w})} + \underbrace{\frac{\alpha}{2} \|\mathbf{w}\|_{2,p}^2 + \beta \sum_{m=1}^{M} \mu_m r_m}_{\Omega(\mathbf{w})}, \tag{3}$$

where $\ell(\mathbf{w}, \phi_{\boldsymbol{\mu}}(\mathbf{x}_i), y_i) = \left| 1 - \left( \langle \mathbf{w}_{y_i}, \phi_{\boldsymbol{\mu}}(\mathbf{x}_i) \rangle - \max_{y \neq y_i} \langle \mathbf{w}_y, \phi_{\boldsymbol{\mu}}(\mathbf{x}_i) \rangle \right) \right|_+$ and $r_m = \sum_{j>\zeta} \lambda_j(\mathbf{K}_m)$ is the tail sum of the eigenvalues of the $m$-th kernel matrix, $m = 1, \ldots, M$.

---

**Algorithm 2** SMSD-MKL

---

**Input:** $\alpha, \beta, \boldsymbol{r}, T$
**Initialize:** $\mathbf{w}^1 = \mathbf{0}, \boldsymbol{\theta}^1 = \mathbf{0}, \boldsymbol{\mu}^1 = \mathbf{1}, q = 2 \log K$
**for** $t = 1$ **to** $T$ **do**
    Sample at random $(\mathbf{x}^t, y^t)$
    Compute the dual weight: $\boldsymbol{\theta}^{t+1} = \boldsymbol{\theta}^t - \partial C(\mathbf{w}^t)$
    $\nu_m^{t+1} = \|\boldsymbol{\theta}_m^{t+1}\| - t\beta r_m, \forall m = 1, \dots, M$
    $\mu_m^{t+1} = \frac{\text{sgn}(\nu_m^{t+1})|\nu_m^{t+1}|^{q-1}}{\alpha\|\boldsymbol{\theta}_m^{t+1}\| |\nu_m^{t+1}|_q^{q-2}}, \forall m = 1, \dots, M$
**end for**

---

Based on the stochastic mirror descent framework for minimization problems in [31, 29], we design a stochastic mirror and sub-gradient descent algorithm, called `SMSD-MKL`, to minimize (3), seen in Algorithm 2.

As shown in the mirror descent algorithm, it maintains two weight vectors: the primal vector $\mathbf{w}$ and the dual vector $\boldsymbol{\theta}$. Meanwhile, the optimization formulation can be divided into two parts: $C(\mathbf{w})$ to update $\boldsymbol{\theta}$ and $\Omega(\mathbf{w})$ to update $\mathbf{w}$ by the gradient of the Fenchel dual of $\Omega$. Actually, the algorithm puts the kernel weight $\boldsymbol{\mu}$ aside when updating $\boldsymbol{\theta}$, but $\boldsymbol{\mu}$ is updated together with $\mathbf{w}$ according to a tricky link function given in Theorem 3.

- For $C(\mathbf{w})$, the algorithm updates the dual vector with the gradient of $C(\mathbf{w})$. Since hinge loss used in $C(\mathbf{w})$ is not differentiable, the algorithm uses sub-gradient of $z^t = \partial \ell(\mathbf{w}^t, \phi_{\boldsymbol{\mu}}(\mathbf{x}^t), y^t)$, where $\partial \ell(\mathbf{w}^t, \phi_{\boldsymbol{\mu}}(\mathbf{x}^t), y^t)$ is the sub-gradient w.r.t $\mathbf{w}^t$.

- For $\Omega(\mathbf{w})$, as in the UFO-MKL [29], the algorithm uses $\mathbf{w} = \nabla\Omega^*(\boldsymbol{\theta})$ to update the primal vector $\mathbf{w}$, of which the calculation has been given in Theorem 3.

The algorithm starts with $\mathbf{w}^1 = \mathbf{0}$, $\boldsymbol{\theta}^1 = \mathbf{0}$ and $\boldsymbol{\mu}^1 = \mathbf{1}$. Especially, the algorithm initializes $q = 2 \log K$ to make the order of generalization reach $\mathcal{O}\left(\frac{(\log K)^{2+1/\log K}}{n}\right)$, according to Theorem 2. In each iteration, the algorithm randomly samples a training example from the train set.

Actually, the algorithm updates real numbers $\|\boldsymbol{\theta}_m^{t+1}\|$, $\nu_m^{t+1}$ and $\mu_m^{t+1}$ in scalar products instead of high-dimensional variables $\mathbf{w}^{t+1}$ and $\boldsymbol{\theta}_m^{t+1}$. The $\|\boldsymbol{\theta}_m^{t+1}\|$ can be calculated in an efficient incremental way by scalar values as following:

$$\|\boldsymbol{\theta}_m^{t+1}\|_2^2 = \|\boldsymbol{\theta}_m^t - z_m^t\|_2^2 = \|\boldsymbol{\theta}_m^t\|_2^2 - 2\boldsymbol{\theta}_m^t \cdot z_m^t + \|z_m^t\|_2^2$$

where $z^t = \partial \ell(\mathbf{w}^t, \phi_{\boldsymbol{\mu}}(\mathbf{x}^t), y^t)$.

**Theorem 3.** *Let* $\boldsymbol{\nu} = \left[\|\boldsymbol{\theta}_1\| - \beta r_1, \dots, \|\boldsymbol{\theta}_M\| - \beta r_M\right]$, *then the component* $m$-*th of* $\nabla\Omega^*(\boldsymbol{\theta})$ *is*

$$\frac{\text{sgn}(\nu_m)\boldsymbol{\theta}_m}{\alpha\|\boldsymbol{\theta}_m\|} \frac{|\nu_m|^{q-1}}{\|\boldsymbol{\nu}\|_q^{q-2}},$$

*where* $\text{sgn}(x)$ *is defined as* $\text{sgn}(x) = 1$ *if* $x > 0$, $\text{sgn}(x) = -1$ *if* $x < 0$ *and* $\text{sgn}(x) \in [-1, +1]$, *if* $x = 0$.

## 6 Experiments

In this section, we compare our proposed `Conv-MKL` (Algorithm 1) and `SMSD-MKL` (Algorithm 2) with 7 popular multi-class classification methods: One-against-One [12], One-against-the-Rest [3], $\ell_1$-norm linear multi-class SVM (LMC) [6], generalized minimal norm problem solver (GMNP) [8], the Multiclass MKL (MC-MKL) with $\ell_1$-norm and $\ell_2$-norm [38] and mixed-norm MKL solved by stochastic gradient descent (UFO-MKL) [29]. Actually, we complete comparison tests via implements in LIBSVM (One-against-One and One-against-the-Rest), the DOGMA library [2] (LMC, GMNP, $\ell_1$-norm and $\ell_2$-norm MC-MKL) and the SHOGUN-6.1.3 [3] (UFO-MKL). We implement our proposed `Conv-MKL` and `SMSD-MKL` algorithms based on UFO-MKL.

Table 1: Comparison of average test accuracies of our `Conv-MKL` and `SMSD-MKL` with the others. We bold the numbers of the best method and underline the numbers of the other methods which are not significantly worse than the best one.

| | Conv-MKL | SMSD-MKL | LMC | One vs. One | One vs. Rest | GMNP | $\ell_1$ MC-MKL | $\ell_2$ MC-MKL | UFO-MKL |
|---|---|---|---|---|---|---|---|---|---|
| plant | 77.14±2.25 | **78.01±2.17** | 70.12±2.96 | 75.83±2.69 | 75.17±2.68 | 75.42±3.64 | 77.60±2.63 | 75.49±2.48 | 76.77±2.42 |
| psortPos | 74.41±3.35 | **76.23±3.39** | 63.85±3.94 | 73.33±4.21 | 71.70±4.89 | 73.55±4.22 | 71.87±4.87 | 70.70±4.89 | 74.56±4.04 |
| psortNeg | 74.07±2.16 | **74.66±1.90** | 57.85±2.49 | 73.74±2.87 | 71.94±2.50 | 74.27±2.51 | 72.83±2.20 | 72.42±2.65 | 73.80±2.26 |
| nonpl | **79.15±1.51** | 78.69±1.58 | 75.16±1.48 | 77.78±1.52 | 77.49±1.53 | 78.35±1.46 | 77.89±1.79 | 77.95±1.64 | 78.07±1.56 |
| sector | 92.83±2.62 | **93.39±0.70** | 93.16±0.66 | 90.61±0.69 | 91.34±0.61 | \ | \ | 92.15±2.57 | 92.60±0.47 |
| segment | 96.79±0.91 | **97.62±0.83** | 95.07±1.11 | 97.08±0.61 | 97.02±0.80 | 96.87±0.80 | 96.98±0.64 | 97.58±0.68 | 97.20±0.82 |
| vehicle | 79.35±2.27 | 77.28±2.78 | 75.61±3.56 | 78.72±1.92 | 79.11±1.94 | 81.57±2.24 | 74.96±2.93 | 76.27±3.15 | 76.92±2.83 |
| vowel | 98.82±1.19 | **98.83±5.57** | 62.32±4.97 | 98.12±1.76 | 98.22±1.83 | 97.04±1.85 | 98.27±1.22 | 97.86±1.75 | 98.22±1.62 |
| wine | **99.63±0.96** | **99.63±0.96** | 97.87±2.80 | 97.24±3.05 | 98.14±3.04 | 97.69±2.43 | 98.61±1.75 | 98.52±1.89 | 99.44±1.13 |
| dna | 96.08±0.83 | **96.30±0.79** | 92.02±1.50 | 95.89±0.56 | 95.61±0.73 | 94.60±0.94 | 96.27±0.68 | 95.06±0.92 | 95.84±0.61 |
| glass | **75.19±5.05** | 73.72±5.80 | 63.95±6.04 | 71.98±5.75 | 70.00±5.75 | 71.24±8.14 | 69.07±8.08 | 74.03±6.41 | 72.46±6.12 |
| iris | 96.67±2.94 | **97.00±2.63** | 88.00±7.82 | 95.93±3.25 | 95.87±3.20 | 95.40±7.34 | 95.40±6.46 | 94.00±7.82 | 95.93±2.88 |
| svmguide2 | 82.69±5.65 | **85.17±3.83** | 81.10±4.15 | 84.79±3.45 | 84.27±3.03 | 81.77±3.45 | 83.16±3.63 | 83.84±4.21 | 82.91±3.09 |
| satimage | 91.64±0.88 | 91.78±0.82 | 84.95±1.15 | 90.67±0.91 | 89.29±0.96 | 89.97±0.81 | 91.86±0.62 | 90.43±1.27 | **91.92±0.83** |

We experiment on 14 publicly available datasets: four of them evaluated in [38] (plant, nonpl, psortPos, and psortNeg) and others from LIBSVM Data. For each dataset, we use the Gaussian kernel $K(\mathbf{x}, \mathbf{x}') = \exp\left(-\|\mathbf{x} - \mathbf{x}'\|_2^2/2\tau\right)$ as our basic kernels, where $\tau \in 2^i, i = -10, -9, \ldots, 9, 10$. For single kernel methods (One vs. One, One vs. Rest and GMNP), we choose the kernel which have the highest performance among basic kernels estimated by 10-folds cross-validation. Meanwhile, we use all basic kernels in MKL methods (`Conv-MKL`, `SMSD-MKL`, $\ell_1$ MC-MKL, $\ell_2$ MC-MKL and UFO-MKL). The regularization parameterized $\alpha \in 2^i, i = -2, \ldots, 12$ in all algorithms and $\zeta \in 2^i, i = 1, 2, \ldots, 4, \beta \in 10^i, i = -4, \ldots, 1$ in `SMSD-MKL` are determined by 10-folds cross-validation on training data. Other parameters in compared algorithms follow the same experimental setting in their papers. For each dataset, we run all methods 50 times with randomly selected 80% for training and 20% for testing, offering an estimate of the statistical significance of differences in performance between methods. All statement of statistical significance in the remainder refer to a 95% level of significance under $t$-test.

The average test accuracies are reported in Table 1. The results show: 1) Our methods `Conv-MKL` and `SMSD-MKL` give best results on nearly all datasets except *vehicle* and *satimage*; 2) `SMSD-MKL` is better than `Conv-MKL` because it wins on 2/3 datasets; 3) Compared with typical MKL methods, our methods get better results over almost all datasets except that only UFO-MKL works slightly better than ours on *satige*; 4) The MKL methods usually work better than the compared single kernel methods (One vs. One, One vs. Rest and GMNP); 5) The kernel classification methods have better performance than the linear classification machine (LMC) on all datasets.

The above results show that the use of the local Rademacher complexity can significantly improve the performance of multi-class multiple kernel learning algorithms, which conforms to our theoretical analysis.

# 7   Conclusion

In this paper, we studied the generalization performance of multi-class classification, and derived a sharper data dependent generalization error bound using the local Rademacher complexity, which is much sharper than existing data-dependent generalization bounds of multi-class classification. Then, we designed two algorithms with statistical guarantees and fast convergence rates: `Conv-MKL` and `SMSD-MKL`. Based on local Rademacher complexity, our analysis can be used as a solid basis for the design of new multi-class kernel learning algorithms.

# Acknowledgments

This work is supported in part by the National Natural Science Foundation of China (No.61703396, No.61673293, No.61602467), the National Key Research and Development Program of China (No.2016YFB1000604), the Science and Technology Project of Beijing (No.Z181100002718004) and the Excellent Talent Introduction of Institute of Information Engineering of CAS (Y7Z0111107).

## Footnotes

[2] Available at http://dogma. sourceforge. net

[3] Available at http://www.shogun-toolbox.org/

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
