[Reviews · NeurIPS 2018]

Reviewer 1



After author response: thank you for your detailed comparison. ====== Proving generalization bounds is one of the most important tasks in theoretical machine learning. The current paper is focusing on multi-class classification where the output cardinality K is larger than 2. The investigation of multi-class classification has a long history (e.g., [1, 7, 12, 14]). The paper claims to highly improve the generalization bound using the known technique of "local rademacher complexity" ([3]). This technique focuses on a smaller class that contains only hypotheses with small variance. The paper claims to improve the generalization bound from K/sqrt{n} to logK^(O(1))/n (where n is the sample size), which seems like an impressive improvement. Of course, this is under the assumption that the "true" hypothesis is with low variance. It is unclear to me why the authors have decided not to reference and compare to the paper "Xu, Chang, et al. "Local rademacher complexity for multi-label learning." IEEE Transactions on Image Processing 25.3 (2016): 1495-1507" which has a similar goal: "We analyze the local Rademacher complexity of empirical risk minimization (ERM)-based multi-label learning algorithms, and in doing so propose a new algorithm for multi-label learning."

Reviewer 2



I read the reply by the authors and I am satisfied by their answer.

Reviewer 3



In this paper the authors propose a new bound for the multi-class classification setting based on the notion of local Rademacher complexity, which is then used to derive two novel multi-class multi-kernel methods. Theoretical guarantees and experimental results are provided for both methods. To the best of my knowledge, this is the first time that local Rademacher complexity is used to derive multi-class learning methods and as such this contribution is quite interesting and promising. As far as I could tell, the mathematical justifications for both the bound and the methods are sound. The paper is well written, even though there are a few typos (e.g lgorithm line 223, of missing ling 130, etc) and awkward phrasings (e.g. sec 2.3). Edit after rebuttal: I think that the authors did a good job at addressing the various questions raised by the reviewers, as such I'm keeping my rating for this paper.